



# Socio-hydrology from the bottom up: A template for agent-based modeling in irrigation systems

Dimitrios Bouziotas[1] and Maurits Ertsen[1,2]

[1]Delft University of Technology, Water Resources Department, Faculty of Civil Engineering and Geosciences, Delft University of Technology, P.O. Box 5048, 2600 GA Delft, The Netherlands

[2]Correspondence to m.w.ertsen@tudelft.nl

**Abstract.** Based on a review of key concepts in agent-based modeling for irrigation systems and coupled human-water systems in general, this study presents a proof of concept of an agent-based model based on the existing Irrigation

Management Game. After the modeling philosophy and main characteristics are outlined, a number of pilot applications are presented and evaluated. Following the evaluation of the results, future steps that could be incorporated in the model are discussed. The proposed template offers a bottom-up approach to socio-hydrological modelling design, as individual agent behavior explicitly co-shapes the response of the water system, which allows the discovery of emergent dynamics and the conditions under which these are produced. The concepts explained and modeled at a proof-of-concept level in this work

serve as a call to the socio-hydrological community to expand its modeling efforts to the agent level.

## 1. Introduction

A paradigm shift is occurring in hydrological modeling practice: starting with the concepts established in Integrated Water Resources Management (H.H.G. Savenije & Van der Zaag, 2008), the call for holistic thinking in models (Sivapalan, 2005) and, more recently, with the introduction of socio-hydrology as a novel scientific field (Di Baldassarre, Viglione, et al.,

2013; Sivapalan et al., 2012; Troy et al, 2015), water scientists are realizing that human agency is neither a part easily omitted nor an externality that can be loosely integrated with physical models. Instead, human agency is an endogenous factor that co-evolves with the same physical system that the same agency is actively shaping (Ertsen, 2010). This realization calls for a need for models to include human agency as an equally important part in coupled human-water systems, thus abandoning the conventional approach in hydrology that regards it a boundary condition or external forcing to pre-existing

physical models (Di Baldassarre, Viglione, et al., 2013).

Among the tools one may utilize to explore human interactions with physical systems, Agent-Based Models (ABM) are a promising choice; originally from the field of Applied Social Sciences (Epstein, 1999; Gilbert, 2008; Squazzoni et al 2014; see Boman and Holm (2005) for a useful overview of issues related to ABM and microsimulations), their affinity in modeling social interactions, as well as their conceptual clarity, seems to fit with the prerequisite to model the human

condition as an equally considered half in the co-evolution of human-water systems. As a computational method, Agent-



Based Modeling has been used to facilitate, through rules-based modeling and simulation, scientific understanding of complex systems dominated by interactions generated at the level of individual units (hereinafter called agents). Applications of Agent-Based Models, starting from the development of dedicated software environments such as SWARM, StarLogo and NetLogo from 1990 onwards, have been extensive and can be found in multiple scientific fields, ranging from economics

and archaeology to ecosystems modeling and spatial planning (Bonabeau, 2002; Grimm, 2005; Heckbert et al., 2010; Matthews et al., 2007; Wilkinson et al 2013; Wurzer et al., 2015). In water systems, ABM can be an attractive methodology to integrate human interaction and has seen applications in cases such as the study of conflicts in water resources (Akhbari & Grigg, 2013; Kock, 2008) and the study of urban water systems in a broader socio-technical context (Baki et al., 2012). ABM has also been used to study irrigation systems, which will be further discussed in the paragraphs below.

While agent-based frameworks have been envisaged to function well in a socio-hydrological context (Ertsen, 2012; Ertsen et al., 2014; Kock, 2008), modeling efforts for coupled human-water systems in hydrology have instead concentrated on another approach that could be described as a lumped, process-based design framework. In these socio-hydrological studies (Di Baldassarre, Kooy, et al., 2013; Elshafei et al., 2014; Giraud et al., 2002; Liu et al., 2015; Srinivasan et al., 2010; Viglione et al., 2014), human actions are interpreted as an aggregate social response to stimuli from the physical system,

with both aspects being explicitly modeled at the scale of emergence of the hydrological phenomenon (e.g. a floodplain, catchment or river basin). The natural (hydrological) system is set, in modeling terms, through physical (hydrological and hydraulic) models; the response of human units (settlements, farmer communities etc.) is then studied as a reactive mechanism to physical model output, such as flooding.

The key difference of the aforementioned approach with the agent-based logic is the viewpoint: the simulation setup in the

20 first case is built with a bird's eye view on socio-physical processes at a lumped scale, whereas the ABM takes the agent perspective, and especially the way it reacts with other agents and the environment, as the basis for modeling (Macal & North, 2006). It thus offers a bottom-up approach, where conclusions about the system characteristics need to be drawn from studying the collective effect of individual decisions. In other words, ABM are built as microscale models[1] (Gustafsson & Sternad, 2010), operational on an agent level, but their study allows conclusions to be drawn at a larger scale, following the

25 process of emergence. The latter is evidently absent when no multi-scaling architecture is included in the model.

This study aims to be a contribution towards the establishment of a bottom-up approach in socio-hydrology, by presenting a template for a signal-based framework for irrigation systems based on agents instead of processes. Leaning closer to the notions of action-oriented than physical modeling, this approach shapes the vital elements of a coupled human-water system, anthropogenic and physical, into a realm of interacting agent units that receive and transmit signals, forming decisions about

30 their transmissions based on the received information. The developed template is created for run-of-river irrigation systems

---

[1] The term microscale refers to the agent level and is not an absolute spatial scale; agents can be human units, groups of people showing collective behavior or other social/corporal structures. The micro/macro antithesis here is used only to highlight that results come from studying the whole (larger scale) by modeling the parts (agent level). For a more theoretical discussion on scales in relation to human agency, see Ertsen (2016).



and is facilitated by simplifying key features based on the set of rules proposed by the Irrigation Management Game (IMG), an example of serious gaming where human participants are assigned the roles of central management agents or local water users in a simplified hypothetical irrigation scheme (Burton, 1989, 1993). The interest, in the studied case, is to establish standards for exploring, based on a set of simple rules:

- the main interactions between agents in an irrigation system.
- the macroscopic effects decisions of individual agents bring on the system.
- if the main dynamics observed through playing the IMG – that, correspondingly, reflect actual irrigation system dynamics (Ertsen, 2012) - can be simulated with the model.

The study first starts by exploring basic concepts, conditions and prerequisites for an agent-based interpretation of the Irrigation Management Game. It then presents a simple proof of concept and discusses its results. Proposed next steps and further research requirements are also highlighted to guide future work.

## 2. On the conceptual modeling approach

### 2.1. Modeling philosophy

Beyond standard modelling guidelines, an agent-based model requires the modeller to consider some unique aspects that stem from the need to rescale to an agent perspective (Bonabeau, 2002; Macal & North, 2006). More specifically, during the design process one needs to: (a.) identify the key agents, (b.) gain an understanding of agent behaviour and identify their relationships and interactions and (c.) build, validate and assess model behaviour by looking at the macro-scale effects. At the same time, any model is by definition a simplification of reality; as such, it is reasonable – and mandatory, according to the principle of parsimony (Blumer et al., 1987) – to make abstractions and retain only the basic dynamics. However, keeping a parsimonious structure should not be misinterpreted as a justification of oversimplification: important characteristics should be retained and core dynamics should be reflected in the results. In the context or irrigation systems, some important aspects to be considered while modeling are:

(a) the main behavioural patterns which characterize human agency in the system, both in terms of the central management and the local actors. What is of importance is to identify and map the types of interaction between different agents of the system and discover behavioural patterns and equilibria among them.

(b) the laws of the physical system and in particular the hydraulics in the canal network, the inherent uncertainty in the provided water resources (particularly important for dry climate cases or run-of-river schemes) and basic crop dynamics (such as seasonality in crop stages, yield response etc.) that, in turn, affect human decision-making.

(c) the interplay of various spatial and temporal scales (Ertsen, 2012; Ertsen et al., 2014). At minimum, characteristic time scales should be outlined and the decision on which scale to model should be clarified and justified to fit modelling and research needs. No discrepancy between the spatial and the temporal dimension should be apparent, as the two scales are inter-related.



## 2.2. The basis – Irrigation Management Game

Having the aforementioned in mind, the proposed template follows the same main features of the original Irrigation Management Game (IMG) (Burton, 1989, 1993), a role-playing exercise in irrigation systems, where a main canal supplies water to eight tertiary land units of 40ha irrigated area each. The irrigation scheme is run-of-river and no reservoirs or other

flow regulating works upstream are apparent, as is the rule in many schemes, especially in developing countries. The participants in the IMG can take the role of either agency staff (water allocators and water distributors, who decide on how to distribute water among the farmers) or farmers (water recipients, who make decisions about the type of crop they will plant, but are also allowed to negotiate and trade water among themselves). The game then unfolds as an exercise in interaction and communication between farmers and the water agency, as well as between farmers themselves. Issues such as the water

allocation policy employed, the yield response to water and the assessment of system performance (in terms of total yield and economic gains) can be then explored within the game framework (Burton, 1993).

Different irrigation games have proven to be good learning contexts for different target groups, including students (Seibert and Vis 2012; Ertsen 2012), irrigation managers (Burton 1989, 1993; Scheer 1996) and farmers (Scheer 1996; Janssen and Anderies 2013; Janssen et al 2012; Janssen et al 2013), as they allow a useful synthesis between real worlds, experimental or

learning setups and associated logistics. Most games are designed to experience – the difficulty of - water sharing, choice making in terms of water use, and the results of strategic actions of players in terms of gains or losses. In the Irrigation Management Game (Burton 1989; 1993 – and the very similar River Basin Game (Lankford et al 2004), differential access to water is an essential element. Upstream users have different options compared to downstream users, as water availability generally decreases when going from upstream to downstream (compare with Janssen et al 2011). A specific feature of the

IMG is that (over)allocation and use of the resource is not expressed indirectly through gains, but rather directly in water being available or not for players. What they do with their water is their choice. As such, the IMG typically includes richer dynamics then most games. At the same time, it does not include all dynamics one can encounter in real irrigation systems. This last feature makes the IMG a highly suitable "compromise" between real life and model world (Ertsen 2012).

Figure 1 shows an overview of the original IMG mapping, where the river diversion and the eight tertiary units are visible.

The game is played in a number of seasons (usually two) and with three crop stages in each season; water input in the upstream river and precipitation on the tertiary units themselves are provided in every crop stage. The original version from Burton (1993) divides each tertiary unit into 10 ha segments, with a distinct crop type able to be planted on each part; this allows for more elaborate profit strategies to farmers, which then have to optimize their received water distribution among the four segments. The game also provides three crop types (rice, maize and soy bean), each with different crop prices, water

requirements, costs of purchase and variable yield response to water functions.

The process of forming an agent-based template based on these rules starts by outlining key agents in the original game, as well as their function and role in the system. In its simplest form, the game is played by 10 players; 8 farmers and 2 water agency operators, one being the allocator (Manager) and one being the real distributor (Gate Controller). Apart from the



players, two trainers, with the roles of Game Controller and Trader, supervise the game. Table 1 summarizes the key characteristics of these agents; with the exception of the trainers' role, who for simpler game settings can be omitted, one may see three basic agent units: the Manager (M) , the Farmer (F) and the Gate Controller (GC). Multiple layers of interaction and information exchange can be identified around these three agents: Managers interact with Gate Controllers to

bring the desired distribution to reality with the canal gates, Farmers interact with the Manager or the Gate Controller to lobby or override the quantities of water given, Farmers interact with other Farmers to trade water. Moreover, each Farmer has to optimize their distribution of water to four segments in relation to corresponding crop selections.

## 2.3. From real to digital game settings

The abstraction process for the template begins with a note on systems complexity and the importance of simplifications in
computational agent networks. Working with a signal-based approach, where every type of agent is an entity that receives and transmits signals, let us assume an irrigation system with $N$ agents, one Distribution Manager, one Gate Controller and $N$-2 Farmers. In principle, each agent in the model has to connect with every other $N$-1 agents in a unique, directional line (arrow signal); as the arrow of information is unique, the signal $N_1 \rightarrow N_2$ is not the same as $N_2 \rightarrow N_1$. This results in $N(N\text{-}1)$ possible unique signals being transmitted among agents at every stage of the simulation. In the case of $N = 10$ (8 farmers and
2 controllers), this results in 90 signals being delivered at every iteration (see Table 2). In the context of the IMG, we may simplify by excluding the signals from the Gate Controller to the Manager and to the Farmers. The reason for this is that the Gate Controller only accepts information (he is commanded by the Manager to distribute water, but can be also lobbied by the Farmers to change his distribution) but does give back information to his "commanders". As there is one Controller, one Manager and $N$-2 farmers, this results in $\mathrm{N(N-1)} - 1 - (\mathrm{N-2}) = \mathrm{N}^2 - 2\mathrm{N} + 1 = (\mathrm{N-1})^2$ signals at every iteration, meaning
that 81 signals are needed per stage for $N$=10. Figure 2 provides an overview of these cases in the form of a network diagram for the case of $N = 5$ agents.

Now, let us assume another irrigation system with the same layout, but with no interactions between farmers (i.e. farmers are not trading water). Knowing that there are $J = N$-2 farmers, the unique signals among farmers are $J (J\text{-}1) = (N\text{-}2)(N\text{-}3)$. The removal of these communication signals gives the new total of $\mathrm{N(N-1)} - 1 - (\mathrm{N-2}) - (\mathrm{N-2})(\mathrm{N-3}) = 3\mathrm{N} - 5$ unique
signals (see also Figure 2). The resulting signals needed per simulation step for a various number of agents $N$ are shown in Table 2. Looking at the mathematical expressions, one may see that the reduction of intra-farmer interactions has resulted in the loss of nonlinearity (the second-degree term $\mathrm{N}^2$) in the number of transmitted signals; the expression in the new case is now linear. This, depending on the number of agents $N$, results in a signal reduction percentage that ranges from 69% (for a simulation where $N$=10), to about 97% for a simulation of $N$=100. Likewise, in a game with $N = 10$ agents and $m = 3$ time
steps, the number of bits of information exchanged over agents is reduced from 243 to 75. Figure 3 provides an overview of the consequences of this abstraction for the case within the IMG context with $N^2$-2$N$+1 as well as the more general case with $N(N\text{-}1)$, showing that an overwhelming advantage in computational time can be gained, especially in the case of multiple



iterations $m$ or multiple agents $N$. These cases might not correspond to the real-world gaming reality of the IMG, but they are theoretically interesting nonetheless, as they allow the exploration of long-term (i.e. steady-state) conditions in irrigation system dynamics (i.e. system states reached after many seasons or once a threshold of farmers is reached (Koutsoyiannis et al., 2003)), a finding that could not be otherwise discovered in the real IMG application. Even in the setting that corresponds

to the real game application ($N$=10), a reduction of approx. 70% in computations is feasible.

With regards to the importance of abstractions indicated by the example above, the following simplifications to the real layout of the IMG are performed at this preliminary approach:

- Firstly, no farmer trading system is considered. The farmers are able to adapt their behaviour based on the water they receive and may interact with the agency (in order to lobby or override centrally planned behaviour) but they do not
have interaction with other farmers. This simplification is made to reduce the degree of computational complexity and amount of dynamic feedbacks per iteration, as explained above.

- Secondly, only one type of crop is allowed to be planted per tertiary unit. This can be considered as a monoculture setting, where all four 10 ha segments bear the same type of crop. This simplification was made to reduce the degree of complexity in farmer decision-making; in contrast to the simpler role of the central managers, the farmers' role is multi-
dimensional, as they have to make decisions on the crop type to plant, on whether or not to override central agency decision, and on the water distribution to four segments per crop stage (so thrice in a season). The monoculture assumption substantially simplifies the former and ignores the latter function, which, when considered, adds considerable complexity and requires a separate module in the code (probably with the use of optimization algorithms to find the optimal distribution per tertiary unit).

With these simplifications in place, the layout is able to explore different central water allocation policies in the eight tertiary units, as well as the ability of farmers to override central agency decisions by modifying gate settings set by the Controller.

## 3. Analysis of the model

### 3.1. Model assumptions

As mentioned earlier, the modeling framework is based on the framework of rules proposed in the Irrigation Management
Game (Burton, 1993; Ertsen, 2012), with slight modifications in prices to allow for the monoculture assumption to be profitable (Table 3). The following assumptions are made at this proof of concept level to facilitate modeling:

- The modeled agents exhibit equal behavior and behave as individualists, i.e. their decision-making is based on their own perception of the modeling reality and not on general system dynamics. They are thus unable to sense lumped values (such as the total water in the river or total profit) and trust their actions only based on their own interest. This
assumption reflects a setting where individualism and competition dynamics are dominant, with a low level of collaboration among agents, thus fitting well with the afore-mentioned removal of intra-farmer trading. It also reflects cases of real human agency (Mueller, 1986) and realties in many large-scale irrigation systems, where farmers would



not have a clear picture of total water flows and distribution logics. This also reflects the principles of agent-based modeling, where simple individual actions shape system behavior (Macal & North, 2006).

- In the original IMG, an individual agent (the Trader) is assigned in charge of deciding crop prices based on the total yield of each crop type. The crop prices may vary moderately, thus reflecting localized market dynamics, where a variation in supply actively reshapes prices. To simplify game settings in this case, a static price system is assumed, with values close to the average price in the IMG (Table 3). While introducing a trader function is simple (linear relationships between system yield and price are provided in the IMG set of rules), it enhances complexity and requires more policies than the ones modeled. The static price system is not unrealistic, since it reflects large-scale market conditions; the market for all products is large enough to sustain stable prices, despite the variation in the production of a small-scale irrigation system.

- The game follows the time-step setting of the original IGM: the basic time step is the season, refined further in three crop stages. This time scale, albeit coarse for operational irrigation system management, is adequate to explore the multi-annual evolution of the system and the response of the farmers to inter-annual hydrological variability. It also offers computational advantages, as water flows can be aggregated to volumes in a seasonal water balance and do not have to be explicitly modeled. The time horizon of the simulation covers any number of seasons (see below).

- The original game settings assume a variable Field Efficiency Factor (FEF), i.e. a soil/crop productivity coefficient, per block of 10ha. As there are no blocks within tertiaries in this case, a uniform FEF of 0.55 is aligned to all tertiary units, so that individual agents will not be handicapped or promoted solely based on the variation of tertiary efficiency.

Besides these assumptions, the crop characteristics (i.e. crop status based on water allocation), as well as the yield response to water functions, are modeled as in the original version (see Burton (1993)). Regarding the modeled agents, an abstraction is made to all agents that are just controllers and do not have a dynamic feedback or play an active role in the system. Based on the characteristics shown at Table 1, the Trainer roles (Tr) can be omitted in a computational setting, replaced by simple calculations within the model. Moreover, the largely mono-directional, controlling role of the Gate Controller (see previous note and Figure 2) can be substituted by an override module within the Farmer agents explained below.

## 3.2. Simulation steps

Focusing on the interplay between Farmer Agents ($F_1, F_2, \ldots, F_n$) and the Management Agency (M), the layout is modeled with the signal philosophy seen in Figure 4. Stochastic elements, in the form of water input as well as modeling tools in agent-based decision making, are explored throughout the framework. Based on the sequence of play seen at the original IMG, the steps of the modeling framework for each season, with *n* Farmers and one Management Agency, are the following (Figure 4):

1.) At the beginning of each season, each Farmer obtains information about crops (prices, water demands, yield potential) and has to decide on the crop type he will plant. The decision is made on probabilistic terms, given an initial set of





probabilities of choice $\{P_c\}$ for each crop type. Uniform triplets of probabilities (see Table 3) are provided as initial conditions for the simulation; these initial "guesses" are scaled based on the profitability and yield potential of each crop.

2.) Following the selection of each crop type, the Farmer agents send demand signals $D_i$ to the Management, asking for a particular quantity of water. Two policies are identified to help with the asked water quantity: the first one (a) is a maximalist

farmer strategy, where every farmer asks for the quantity that will sustain good condition (G) for the chosen crop type throughout the whole season. The second one (b) is more moderate, and aims for a seasonal water quantity that will retain the crop in the condition between Medium (M) and Good (G). The Manager then collects all signals, along with information on the incoming river water and decides on the distribution of the influx $\{\boldsymbol{\eta}\}=\{\eta_1,\eta_2,...,\eta_n\}$ to the n farmers. In a similar way to the farmers, the decision of the manager can be based on two policies; the first (a) is an equity policy, where each farmer

receives an amount proportional to his irrigated area $A_i$ (equal per farmer in the studied case), regardless of his signal (Equation (1)). The second one (b) is a water allocation policy based on the proportionality of the received demand signals (Equation (2)). The result, in any case, is the distributed water $Q_{distr,i}$ in each Farmer $i$.

$$\eta_i = \frac{Q_{distr,i}}{Q_{river}} = \frac{A_i}{\sum\limits_{i=1}^{n} A_i} \quad (1)$$

$$\eta_i = \frac{Q_{distr,i}}{Q_{river}} = \frac{D_i}{\sum\limits_{i=1}^{n} D_i} \quad (2)$$

3.) At a third stage, the distributed water signals are received by each farmer consecutively (starting with $F_1$), which then has to decide on whether he will accept this amount or not. In the context of the model, water quantities are considered another type of signal; the notable difference is that the signal, in that case, represents a physical quantity and is thus subject to the conservation of mass. To calculate the amount each agent takes, the following is assumed: each agent $i$ has requested an amount $D_i = Q_{dem,i}$ but is given $Q_{distr,i}$, which has to be taken from the river volume at the entry node of the tertiary unit,

$Q_{riverup,i}$. (see Figure 4). If the quantity $Q_{distr,i} + Q_{rain,i}$, where $Q_{rain,i}$ is the rainfall within the tertiary unit, does not satisfy $Q_{dem,i}$, the agent is given a choice: to accept the lesser quantity given by the Manager or override it. Overriding, in that sense, is an aggregate action-decision that includes real-game actions such as bribing the Gate Controller, bribing the central Manager (to deviate from the ideal distribution) or resetting the entry gate of his tertiary unit; in the modeling domain, it is represented with an override probability $P_{ovr}$ (Table 3), equal in each agent. When an agent attempts an override and is successful, he

will retain as $Q_{kept,i}$ the quantity $Q_{dem,i}$ instead of $Q_{distr,i}$ from the upstream quantity $Q_{riverup,i}$ available to him, provided that there is the actual quantity he wishes to receive. A successful override leads to agent $i$ leaving a smaller quantity of water at the downstream part of the irrigation network $Q_{riverup,i+1}$, which is also what the next agent receives as his available water signal (Figure 4, lower left panel). A sink quantity $Q_{sink}$ is also set in case of a year with excessive rain, enough to cover the





demand and also provide a surplus which has to be dumped locally. Local (agent) and global mass balance checks, in the form of Equation (3):

$$\text{In} - \text{Out} = \text{Storage} \Rightarrow \left( \sum_{i=1}^{n} Q_{distr,i} + n \cdot Q_{rain,i} \right) - Q_{riverout} = \left( \sum_{i=1}^{n} Q_{kept,i} + \sum_{i=1}^{n} Q_{sink,i} \right) \qquad (3)$$

ensure that the water mass conservation law applies to the groups of discharge signals exchanged between agents and to the

system as a whole. Other types of losses (evaporation, soil infiltration) are, as in the original version of the IMG, not considered. Note that Figure 4 implies that successful overrides are not directly communicated to downstream users – these farmers may only realize it implicitly, through observed losses, which is (again) very similar to situations in many large-scale irrigation systems.

4.) As a last step, the total yield and profit of each Farmer $i$ is calculated at the end of each season. With the addition of the

costs, the total revenue is found; in the case of farmer loss, a memory signal, noted with red color in Figure 4, is generated that feeds the farmer agent in the next time step $t+1$. This signal creates the memory that something went wrong and led to financial losses for the farmer. This leads to a change in the probabilities of choice $\{Pc\}_{t+1}$ for that particular agent and for the upcoming season, thus reflecting more conservative/safe choices in terms of water demand and economic exposure. Likewise, the memory of a profitable season shifts the probabilities of choice towards more profitable (and water-

demanding) crop types for the next season. The underlying assumption with this approach is that individual financial profitability is the main consideration for agents to change unsatisfactory or to sustain successful behavior. This seems reasonable, as a financial sign is a substantially more explicit call for change (or preservation) in agent behavior than, for instance, growing awareness of the behavior of upstream users. To what extent memories of past negative events can be considered as a key factor in provoking change (see Di Baldassarre, Viglione, et al., 2013) needs further study, but the

importance of memory is clear.

Once the afore-mentioned structure is prepared, the model can be forced by hydrological input, i.e. seasonal river and rainfall volumes, further distributed across crop stages. For reasons of consistency, the original IMG water input settings are used; incoming river water $Q_{riverin}$, as well as an equal volume of rainfall per tertiary unit $Q_{rain,i}$ , is given. Unlike the IMG, however, a stochastic approach to generate water input is employed. Firstly, the statistical properties of river $X$ and tertiary

rainfall $Y$ volumes are calculated from the sample given through the real-game Water Cards (Table 4), assuming a Gaussian distribution, which seems reasonable for aggregate water quantities at seasonal time scales. During this process, the distribution percentages of water input per crop stage were found to be constant and are given in Table 4. Secondly, stochastic river and rainfall water quantities are generated by drafting from the random variables $X,Y$ with the use of the following equation:

$$\left. \begin{array}{l} X = \mu_X + \sigma_X \cdot Z_1 \\ Y = \mu_Y + \sigma_Y \cdot \left( Z_1 \rho + Z_2 \sqrt{1-\rho^2} \right) \end{array} \right\} \qquad (4)$$

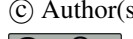



where $Z_1$ and $Z_2$ are independent random variables with the standard normal distribution and $\rho=corr\{X,Y\}$ is the cross-correlation coefficient between $X$ and $Y$. The use of Eq. (4) allows cross-correlation, which is very high for the considered data (Table 4), to be retained in the stochastically generated time-series (Kalos & Whitlock, 2008).

The use of stochastic input has a strong advantage; instead of the 10 originally available water seasons, one may now do simulations with any duration in the water input. This is an obvious advantage of the model versus the reality, as scenarios can be now formed for large numbers of seasons, while a real game is constrained usually in two (Burton, 1993), thus allowing larger output data samples to be collected. In the studied case, long sets of time-series (with a length of 500 seasons) are generated to allow for steady-state system dynamics to develop. The framework also allows for games with an arbitrary number of farmers; however, in order to stay in an analogous setting as the original IMG (Figure 1), in this paper

games with $n=8$ farmers are simulated.

## 4. Results

Following the formulation of the framework described above, a number of simulations are run to explore the dynamics of the system. The results can be interpreted in two levels; firstly, on the level of individual agent performance, i.e. crop selection, water demand, water use, profitability, and secondly on the level of the total system, i.e. total yield and profit. The effect of

different decision strategies (policy types, both in the farmers and in the management – discussed above) is also explored, as the model allows for a total of four different farmer-manager policy combinations. The results can be then compared with the ones obtained from real-world applications of the IMG (Burton, 1989; Ertsen, 2012).

The results obtained on the agent scale, in terms of crop type selection and output performance, can be seen in Figure 5 and Figure 6. The first figure shows the percentage of selected crop types during the 500 simulated seasons for all agents, while

the second figure shows the statistical properties of the agents' performance during these seasons in the form of box plots. Both figures show clear upstream-downstream patterns in the model, with the upstream agents securing more water than the ones downstream and thus producing larger and more consistent profits. This is also reflected in the crop type choices, as upstream agents stick with profitable crop choices (with the rice being more boosted as an option than its initial probability of choice $P_{c,rice}$ given in Table 3), while downstream agents increasingly choose soya as a less water-demanding option.

What is of interest is that, even though that the probability of override is relatively low per agent (see Table 3), downstream farmer agents systematically feel the impact of it, as multiple upstream users try – each one individually - to intervene per season to secure their interests in water. While the first four Farmers experience approximately the same distribution in water use and profits, the other four Farmers experience an increase in the variability of water retained in the canal, and progressively a significant loss in its mean value. This leads users 7 and 8 on to systematic financial unsustainability, as seen

from the middle panel of Figure 6. In terms of retained water, these farmers receive the amount of water farmers 1 and 2 regularly see only 10% of the time ; this percentage is of course sensitive to the model settings, mainly the water input and



the override capability of the users. We did not explore this sensitivity further for this paper, as we present the model as a proof-of-concept, but are planning to explore this issue further.

Evidently, downstream users' profitability is constrained by water scarcity at these positions. However, when the profits are divided to the water used (Figure 6 – lower panel), the results show that downstream users demonstrate the capacity to

generate more profit with their scarce means; user 7, for instance, obtains a better profit ratio than the rest of the agents 10% of the time and, on rarer occasions, much better results than the upstream users. User 8, while being heavily constrained by water scarcity, manages to get the highest profit rations, albeit with high rarity. This behavior is not consistent, but it is an interesting finding that resembles real-game results, with downstream farmers being able to get a higher "crop per drop" (Ertsen, 2012); this behavior can be considered an application of the "scarcity-triggers-wisdom" principle in water

management (Koutsoyiannis, 2014). The inconsistency implies that agents are not  adapting to a high crop to drop ratio strategy systematically; however, the potential to pursue it is inherent in the system dynamics. We are planning to study this issue further, amongst others by refining agent complexity in crop selection, especially by adding a division of fields within each agent unit instead of the monoculture assumption, so that the agents will be able of trying $3^4$ combinations instead of $3^1$.

Since the model setup permits four different policies (two for the Farmers' water demand requests and two for the Manager's

water allocation), the impact of different policies on an agent and system level can be compared. Policies can be considered different scenarios in a long simulation[2], thus leading to different outcomes in terms of profits and water uses. Figure 7 shows the impact of the four different manager-farmer policy combinations in seasonally averaged profits, both in the agent and in the system scale; the first index (*a* or *b*), denotes the maximum or moderate water demands policy of the Farmer agent correspondingly. Likewise, the second index (again *a* or *b*), denotes the Manager water allocation policy of either equity

(Equation (1)) or distribution proportional to request (Equation (2)) correspondingly.

Both panels of Figure 7 show some interesting results: when aggressive supplier and demand policies align (as in the case of (ab), i.e. maximum farmer demands and division of water proportional to their request), the results are much better for upstream users at the cost of tail-enders and, eventually, a marginally better performance (~7%) for the system is achieved as well. This may seem counterintuitive, but it is reasonable in case the amount of extra water supplied at the tail-enders makes

a big difference in the crop health and the yield capacity of the front-runners. Besides this aggressive combination, no other policy leads to increased system performance, as they all lead to approximately the same system profit. However, the spatial distribution of averaged profits differs substantially. By enabling the policy of more moderate demands in all farmer agents, the systematic losses of tail-end users become profits, while the overall system performance remains the same. This implies that, by employing more equitable policies and restraining demands, the management might be able to lessen harsh impacts

to tail-end users, while not suffering from a less productive irrigation system as a whole.

---

[2] The real IMG changes the management rules and scenarios per season (for instance, season 2 allows farmers to override the decision of management while season 1 does not permit that freedom), so that players may explore the difference in outcomes in a single game. In the context of a digital model, this feature does not make sense; instead, each difference policy combination can be a setup to a different long simulation, to allow for stationary conditions to develop and provide consistent statistical samples.





## 5. Discussion

Undoubtedly, the presented work is not a complete recipe for agent-based applications in real irrigation systems; there is still a high degree of abstraction, both in the level of agent complexity and physical process representation. For the sake of computational efficiency, simplifications were made that might predefine agents' actions too much and as such reduce

possibilities for emerging dynamics. At the same time, the setting of the template is based on the imaginary settings of a real role-playing game, the Irrigation Management Game, instead of a real-world system, which allows for reaching a useful compromise between complexity and abstraction. Having said that, the aim of this paper was not to adhere to realism, but to explore the essential elements of agent-based modeling in a simple socio-hydrological context, without succumbing to the perplexing complexity of real irrigation systems (Ertsen, 2010). Having said this, an exhaustive analysis of underlying

system dynamics or a presentation of detailed modeling results with multiple rules and sensitivity analyses would be out of scope. Instead, the aim is to serve as a scoping study that: (a.) introduces essential concepts and challenges of agent-based modeling for socio-hydrological applications, (b.) balances between the presentation of these key conceptual matters and a set of simple proof-of-concept applications.

The choice of the IMG as a rules base seems to fit these goals, as it provides a coherent set of simple rules and input data, as

well as a coarse physical environment where agent behavior is prioritized over meticulousness in hydrological process detail. Moreover, it is argued that the use of this 'toy-model-of-a-toy-model' approach has the following additional advantages:

- The existence of the IMG as a parent structure, with ample data available from multiple real gaming applications (Burton, 1989, 1993; Ertsen, 2012), enables the proposed framework to be validated much more easily than a real system. Serious gaming and real role playing has been employed before as a validation scheme for agent-based models,

and has been found to be an effective way of gaining insights on the mechanisms of human agency and the dynamics of coupled social-physical systems in general (Ligtenberg et al., 2010). The stochastic architecture of the template facilitates validation even further, as the rules that drive decision-making are not arbitrary parameters but tangible probabilities. The probabilities of crop type selection $\{P_c\}$ and the probability of override $P_{ovr}$ can be directly observed and measured from real game applications, thus reflecting the behavior of actual human agents. Easier validation allows

for the exploration of ways to integrate human agency in socio-hydrological models without being hindered by data scarcity or parameterizations heavily dependent on qualitative variables like emotion and trust (Batty & Torrens, 2001; Ligtenberg et al., 2010).

- A toy model facilitates a fusion between two agent environments, for instance games where human agents can share a game with AI in a Web-based setting. This allows for further promoting the IMG as an awareness tool for involved

stakeholders in irrigation systems worldwide - a single browser game is far more easily implemented than a daily session with 8 human players. At the same time, it leads to a broader data pool in water-oriented human agency, which forms a fertile soil for more elaboration in modeling socio-hydrological systems.



In light of these points, the proposal for future studies is to further integrate these two gaming environments, by modifying the initial IMG settings or by adding complexity to the artificial agent scheme when needed. This will bring the proposed template closer to the approach of "companion modeling", where models are developed as an iterative process involving real-world counterparts of the agents. This approach directly embeds model development within the social process of policy

implementation (Barreteau et al., 2001; Ligtenberg et al., 2010), while maximising stakeholder participation in both modeling and policy development (see Janssen and Anderies 2013). Strategies of integration between these two environments need not be elaborate; for instance, a simple questionnaire in the IMG players, asking questions about how and why they made key decisions such as crop selection or overriding management decisions, will shed much light on the reasons for certain choices, allowing extrapolating specifics of human agency and thus refining the template setup.

Beyond this more general integration, increased realism relevant for irrigation conditions can be directly implemented on the model to lessen the degrees of abstraction and provide more faithful physical process representation – without resulting in impossible computational demands. The following main points for such improvements are outlined as follows:

- A key improvement of the model will be to adapt agents' crop selection, by adding a tertiary division (poly-culture instead of monoculture) and a mechanism to scan and find optimal crop selection strategies, possibly fueled by the

success or failure of past schemes and division strategies observed in real games. As described before, this will considerably increase complexity (by a factor of $3^4/3^1 = 27$) and will be non-trivial, as it will also require to solve the optimal water allocation problem, i.e. the way users prioritise certain tertiaries over other ones when water does not suffice. The complexity points to a Dynamic Programming-Linear Programming (DP-LP) problem with incremental solutions; even in the case of a good formulation, one must be weary not to model the *perfect* tertiary division (coming

out of a computationally optimal solution) but merely the *imperfect but progressively better* one chosen by the accumulated wisdom of agents, which is dependent on the location in the irrigation system (Ertsen, 2012).

- Future tangible improvements can be achieved by formulating water trade mechanisms among Farmers (another considerable increase to model complexity according to Figure 2) as well as refining the mechanisms that alter the probabilities used for decision-making. The latter could be investigated by observations and questionnaires from real-

life human agents. Interesting areas of research emerge in case collected data allow for more elaboration on the architecture of the decision-making rules; for instance, agents could base their probabilistic reasoning based on a linguistic approach that utilizes fuzzy logic, with the use of Fuzzy Implementation Systems (Bouziotas et al., 2014; Rozos et al., 2011) . This brings the agent decision-making process closer to the "fuzzier", real human reasoning (Li et al., 2004; Mantelas et al., 2012).

•   In principle, more detail on hydraulics is desirable, given the many observations in actual irrigation systems that show the importance of flow regimes and their actual behavior (in contrast to the water balances that are typically used). Such additional detail can be readily implemented in the model setup without deviations from the logic of signals presented above. Considering that water is a type of signal, including more physical process detail means that the signal needs not be an aggregate seasonal volume but, instead, an (averaged) daily hydrograph of incoming discharge at the upstream



node. In case of large irrigation systems, this hydrograph can be altered with the use of another type of agent in the model – the Canal Agent – that has the sole purpose of routing, i.e. accepting water signals from upstream positions and re-shaping them, based on hydraulics, for the downstream users. The exact behavior of the Canal Agents could be assessed with the use of dedicated routing physical models. In this case, overriding could be also elaborated as the

interaction between the upstream Farmer and the Canal Agent. Figure 8 displays this concept, based on the schematization of Figure 4: part (i) presents the initial volume interaction between Farmers and river volumes already modeled, while (ii) presents the case where canal agents intervene to "distort" the signal.  Evidently, this will bring the modeled time step to much finer scale than seasons, thus leading to significantly increased computational times, but will allow a refinement to Farmer agency that includes feedbacks observable only on operational timescales (Ertsen, 2012).

Interesting enough, including hydraulics through Canal Agents is not just a feasible modelling choice, but also a computational version of the realization that agency is not restricted to humans alone (Ertsen 2016; see Latour 1996; 2013).

- In general, we would be interested to study the co-evolutionary transformation of practices and arrangements in irrigation systems, to examine how humans and water together created new, intermingled forms and processes in the

social practice of irrigation. As a routinized set of behaviours consisting of elements that are interconnected: bodily activities, mental activities, 'things' (artefacts) and their use, background knowledge, know-how and states of emotion (Winiwarter et al 2013), social practices typically include human agents and material objects. In such actor-networks, (sets of) actions are employed to realize a number of conditions to start and/or manipulate a process. In irrigation (and water and landscape management more generally) we encounter a large set of possible actions: opening sluices,

modelling, maintenance, etcetera. Through their agency, human actors are continuously (re)shaping their relevant network, linking themselves with other human agents through (elements in) the irrigation system. Modelling along the lines we sketched above could develop storylines on the level of daily actions and routines, to discover the underlying mechanisms of these storylines. These mechanisms could be of the type "action - result - judgement/perception - (re)action" (compare with Steenbeek and Van Geert 2013). This clear distinction between action and judgement may be

an answer to the problem of assigning agency to individual actors. In the analysis, agents will be offered a spectrum of possible actions. From these, an agent selects possible actions based on his/her current perception of the irrigation context. With the perception that can be changed in the analysis, actions themselves and their results have clearly defined physical boundaries, allowing for clear validation of model results, even in cases without measurements (like in archaeology or future studies).

## 6. Conclusions

In this study, an agent-based template for a stochastic, bottom-up socio-hydrological application is presented, following the rules of the Irrigation Management Game (IMG), a serious gaming environment for irrigation systems. Despite the simplicity



in mechanics and the limitations in the demonstrated pilot applications, emergent dynamics can be observed: clear upstream-downstream patterns are formulated in the irrigation system, with headlining upstream agents getting a clear advantage over tail-enders' water and profits ; however, this does not necessarily mean that downstream users do not have the potential to survive in the long-term (if the right policies are implemented) or transform their scarcity into managerial wisdom. The

5 findings resemble real game applications and, in a broader sense, occurring mechanics in real-world irrigation systems (Ertsen, 2010, 2012).

While the template has limited capabilities in real-world applications, it can be used as an exploration tool as well as a basis for more elaborate human agency in irrigation systems and, more generally, in systems where water needs to serve the conflicting needs of multiple water users. The proof-of-concept application shows the potential to mimic real game and

10 system findings, despite its much simpler character; the next step should be to closely integrate this template with the real IMG and carefully add complexity, especially in agent decision-making and the conflict-cooperation interplay between Farmer agents. It is argued that the template architecture shown here provides a solid base for expansion, as it exhibits clarity, simplicity, potential for actual calibration, parsimony in parameters and can be readily paired with a real human agency environment. At the same time, the simplicity of the template is readily used as a basis for demonstrating core agent-

15 based modeling concepts in an applied context, beyond theoretical integration analysis and wishful thinking about future socio-hydrological applications.

Beyond this first exploratory level of application, a future version of the model is envisaged to be: (a.) calibrated against a combination of observed probabilities "in the field" (i.e. in real game settings) and (b.) further refined based on questionnaire information provided by real human agents in the original IMG. The first improvement will be used to validate the final

model probabilities, while the latter will be used to reshape, in a more accurate basis, the rules behind agent decision-making and help clarify human agency in real-world irrigation systems.

The authors are hopeful that the concepts analysed in this work serve as a call to the socio-hydrological community to explore human interactions on the agent level and not only on the top-down, lumped scale, as seen in the first truly integrated socio-hydrological modeling applications (Blair & Buytaert, 2015; Viglione et al., 2014). The simultaneous deployment of

25 and systematic comparison between both top-down and bottom-up approaches will spark scientific debate and thus significantly enrich the emerging study of socio-hydrology as the intricate co-existence between people and water, in different physical systems and across multiple scales.

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





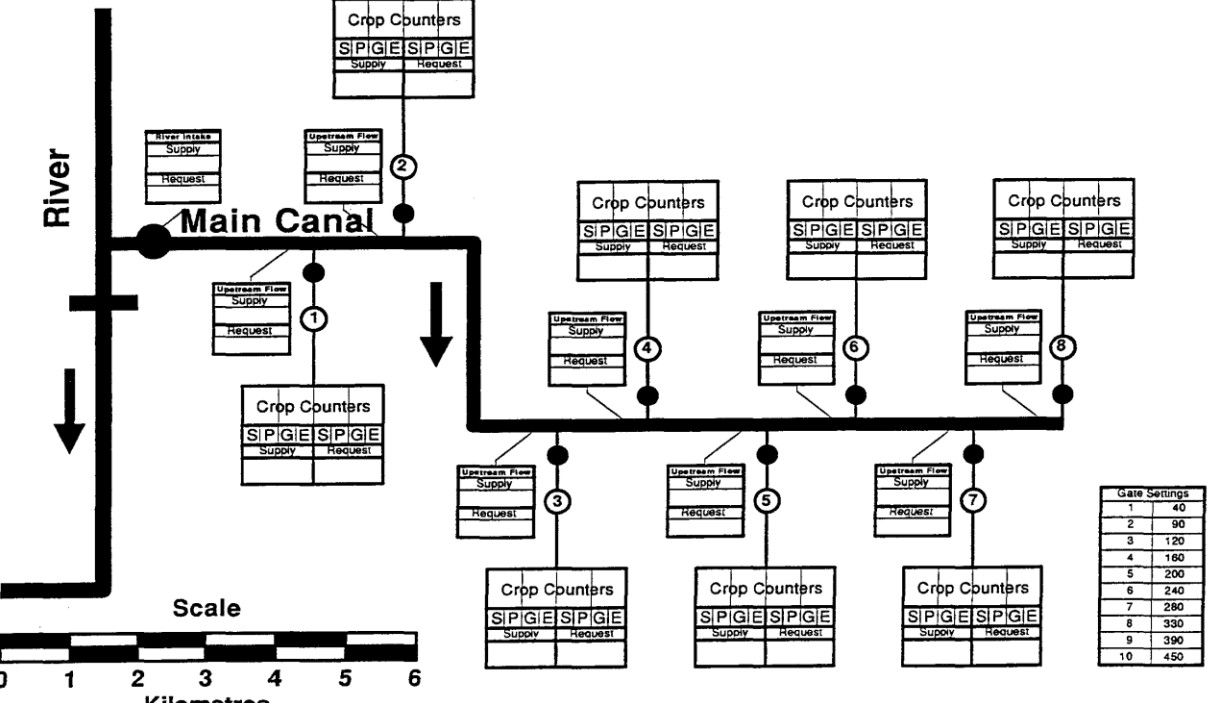

**Figure 1: The irrigation area map, as seen in the original IMG (Source: Burton, 1993).**

**Table 1: Agent typology and function in the IMG.**

| Abbreviation | Name of agent | Function / Role | Comments |
|---|---|---|---|
| IO - M | Irrigation Official - Manager | distributes river water (IO to GC) | Gives command to IO-GC |
| TM | Tertiary Manager (Farmer) | · gets water (distributed from IO-CIA, rainfall)<br>· plants crops in tertiaries<br>· distributes water among tertiaries<br>· negotiates/exerts pressure for more water (TM to IO)<br>· trades water with other farmers (TM to TM) | |
| IO-GC | Irrigation Official – Gate Controller | applies, in a serial manner (1 to 8), the gates according to IO-CIA distribution | Can be bribed or overriden if deviation from the rules set by IO-M is permitted. |
| Tr - GC | Trainer - Game Controller | supervises the gaming process, provides information on the rules, reveals Water Cards | Not a direct game participant |
| Tr - T | Trainer - Trader | sets the prices of each crop type | Not an elaborate agent in non-dynamic pricing systems. |





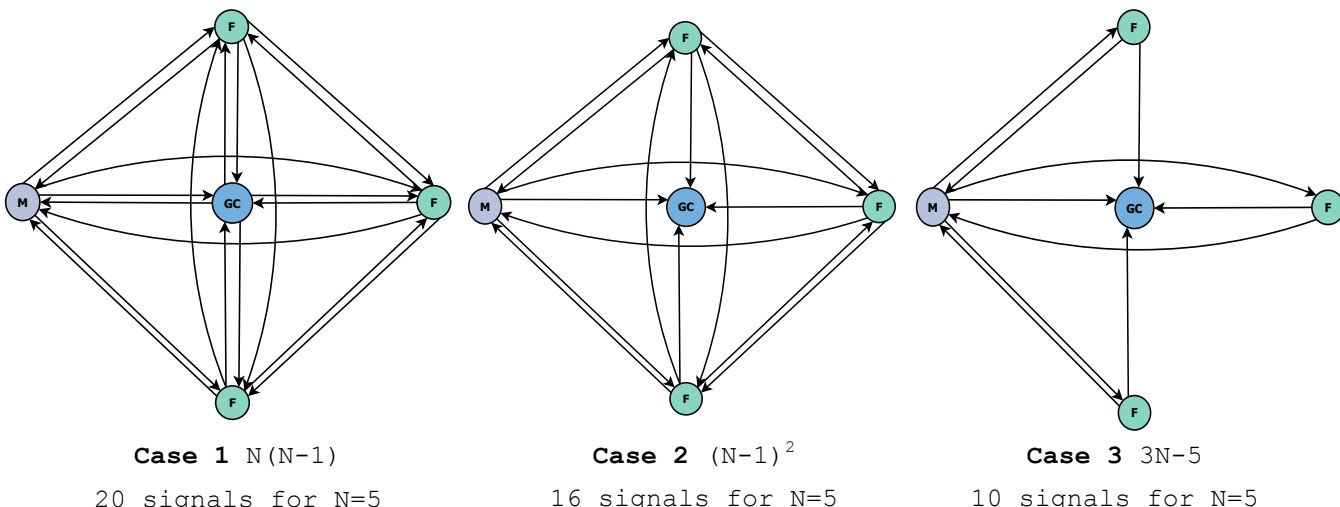

**Figure 2: Network graphs showing the exchange of information for the three cases in the example.**

**Table 2: Total number of signals and computational reduction achieved for the three discussed cases of an agent network that simulates the IGM.**

| | Case 1: General | Case 2: General – Controller Feedback | Case 3: General - Controller feedback - Farmers | Computational Reduction a (%) |
|---|---|---|---|---|
| N | $N(N-1)$ | $N(N-1)-1-(N-2)$ $= (N-1)^2$ | $N(N-1)-1-(N-2)-(N-2)(N-3)$ $= 3N-5$ | $a=1-(3N-5)/(N-1)^2$ |
| 3 | 6 | 4 | 4 | 0.00 |
| 4 | 12 | 9 | 7 | 22.22 |
| 5 | 20 | 16 | 10 | 37.50 |
| 6 | 30 | 25 | 13 | 48.00 |
| 7 | 42 | 36 | 16 | 55.56 |
| 8 | 56 | 49 | 19 | 61.22 |
| 9 | 72 | 64 | 22 | 65.63 |
| 10 | 90 | 81 | 25 | 69.14 |
| 20 | 380 | 361 | 55 | 84.76 |
| 30 | 870 | 841 | 85 | 89.89 |
| 40 | 1560 | 1521 | 115 | 92.44 |
| 50 | 2450 | 2401 | 145 | 93.96 |




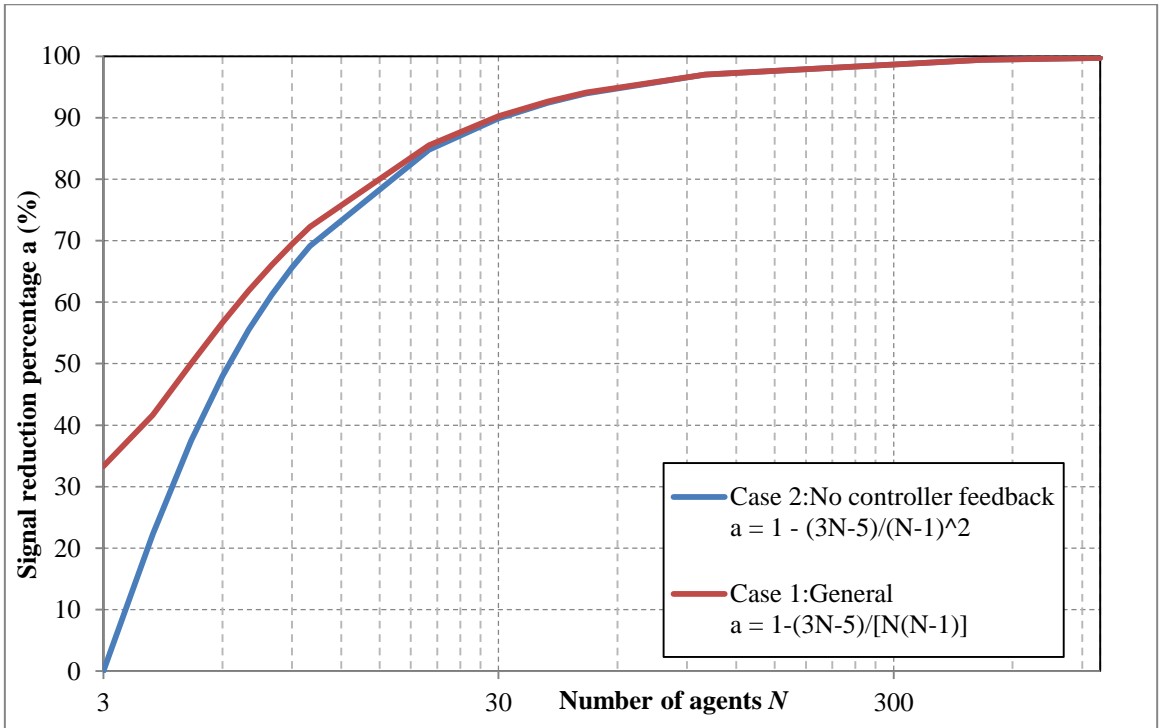

**Figure 3: The signal percentage reduction achieved by simplifying farmer interactions.**



**Figure 4: Flowchart of the proposed agent-based irrigation model. The highlighted bottom left panel provides more details on the Farmer interaction with water input.**





**Table 3: Model parameters and initial conditions.**

| Parameters | | | | |
|---|---|---|---|---|
| Number of Farmers | 8 | | | |
| Field Eff. Factor | 0.55 | | | |
| Subsistence cost [$10^3$ Rp/Farmer/Season] | 6.00 | | | |
| Crop Characteristics | Cost [$10^3$ Rp/10 ha] | Return [$10^3$ Rp/t] | Max Yield [t/10 ha] | Max Profit [$10^3$ Rp/10 ha] |
| Rice | 2.00 | 0.55 | 30 | 16.50 |
| Maize | 1.00 | 0.50 | 20 | 10.00 |
| Soya | 0.80 | 1.25 | 6 | 7.50 |
| **Initial Conditions** | | | | |
| Prob. Of override $P_{ovr}$ | *0.3* | | | |
| | Rice | Maize | Soya | |
| Prob. of crop choice {$\mathbf{P_c}$} | 0.55 | 0.40 | 0.05 | |

**Table 4: Statistical properties of the hydrological input in the IMG.**

| River Input X (Water Units) | | | |
|---|---|---|---|
| Mean Value | Standard Deviation | Distr. Perc. per crop stage | |
| $\mu_X$ | $\sigma_X$ | I | 0.315 |
| 3191 | 725.95 | II | 0.460 |
| | | III | 0.225 |
| **Rainfall Input Y (Water Units/tertiary unit)** | | | |
| Mean Value | Standard Deviation | Distr. Perc. per crop stage | |
| $\mu_Y$ | $\sigma_Y$ | I | 0.283 |
| 130 | 30.82 | II | 0.470 |
| | | III | 0.247 |
| **Correlation ρ:** | 0.98 | | |





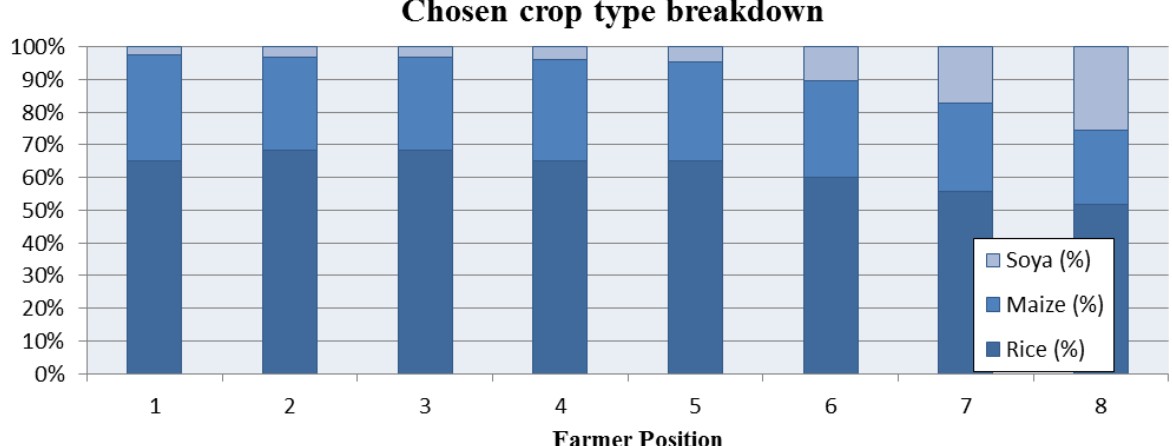

**Figure 5: Crop type selection in the simulated agents.**







**Figure 6: Indicative results of a long (500-season) simulation.**



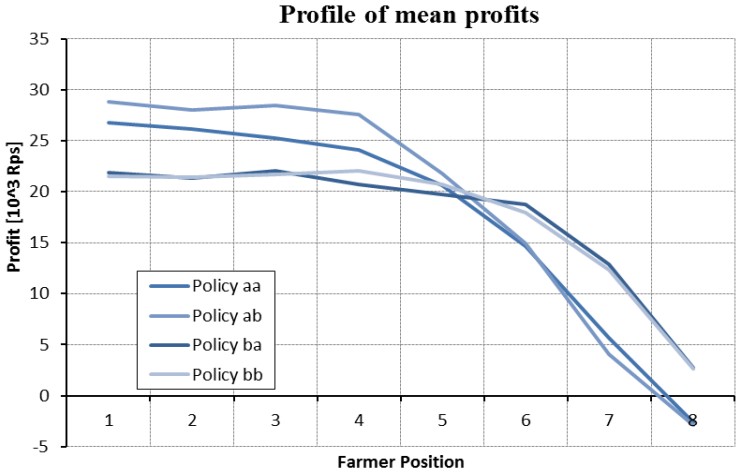

**Figure 7: The impact of different management-farmer policies in mean profits.**

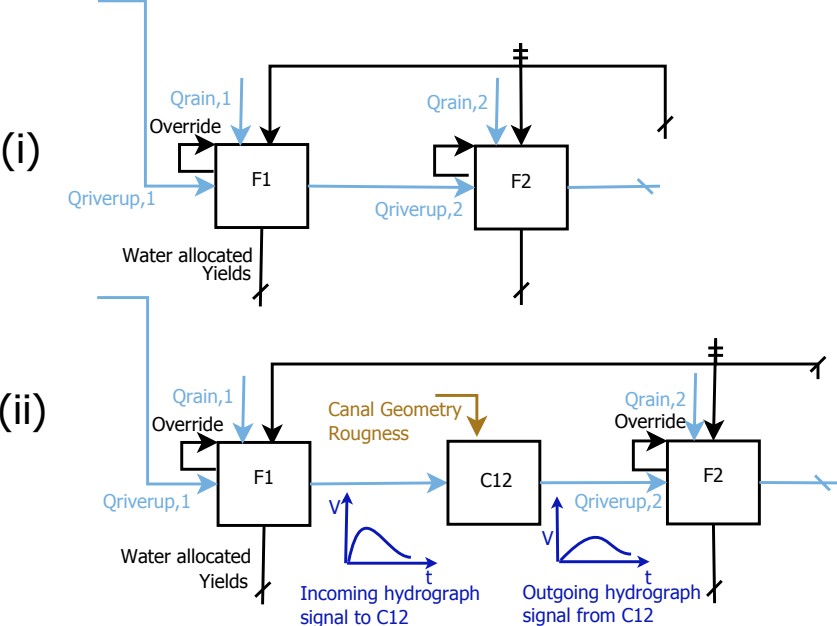

**Figure 8: Elaboration in hydraulic detail through a signal-based approach.**

