# Peer review of "Socio-hydrology from the bottom up: A template for agent-based modeling in irrigation systems"

_Hydrology and Earth System Sciences, 2017_

## Referee Comment (RC1) · Anonymous Referee #1 · 8 Apr 2017

I read this paper with a lot of interest due to the catchy title. However, I came away somewhat disappointed in that it did not deliver on its promise of "socio-hydrology from the bottom-up".

I am not an expert in agent-based modeling, although it is becoming more common in hydrology and water resources applications. This paper is not the first application of its kind. I have seen several in the literature, some even in a socio-hydrologic context. On that point, as an aside, the authors should include at least a brief literature review of ABM applications in this field for completeness.

However, I reserve most of my comments for the topic of the title: socio-hydrology from the bottom-up. I do agree with the authors that the bottom-up approach that includes

[Figure]

ABM has potential to make a contribution to socio-hydrology.

However, I am not clear what this paper contributes to socio-hydrology in a fundamental way. Of course, the paper presents the mechanics of constructing an ABM, and the necessary details of the application to a particular place for a particular problem (irrigation). But what is the purpose of the ABM, what was found, or what was discovered from the modeling. What were the authors hoping to achieve?

I know ABMs have the capability to do up-scaling, i.e., the aggregate behavior of a population of interacting agents. In typical scaling, if everything works well, then ABM must reveal some kind of emergent behavior. The emergent behavior would be a community response, so some aspects of individual agent behavior are lost or left behind, and some macro-scale feature emerges through. I was looking for expressions of such emergent behavior: if they were there they were not brought out well in the presentation. Also for it to make a contribution to socio-hydrology, one needs to highlight the "socio" of the socio-hydrology of the system. This too was not brought out well. The outcomes of the modeling was somewhat underwhelming.

So, at a minimum I expect the paper presentation to be revised to make the goals of the study clearer, and to articulate more clearly and concretely what was learned or discovered through the modeling. Of course, I expect the outcomes to be place-based, and so there needs to be a discussion on what aspects of the place get reflected in the collective community behavior, and in what way they show up in the outcomes. I guess one can elucidate this through doing sensitivity analyses: I admit I am not an expert on the mechanics and will leave it to the authors about how to do it.

Having said that, I want to also raise a philosophical point in respect of the bottom-up and top-down approaches to socio-hydrologic modeling. The authors critique the lumped, top-down approaches more prevalent in the literature. I do agree these models face major challenges, the most difficult of which is about how to simulate collective human-social behavior at the scale of a river basin in the absence of appropriate observations.

Presently the functional forms of these behaviors are arbitrarily assumed, and then validated through calibration of their parameters using observations gross behavior. What social factors (wealth, political systems, institutions, norms and values) impact the behaviors are not well understood.

Surely, Agent Based Models can potentially help – however, this paper only pays lip service to this topic. If the authors want to make serious claims about the potential of ABMs to be turned into socio-hydrology models, then a lot more must be done – the social factors must be highlighted, and the ABM simulations must be designed very carefully to look for emergent dynamics that can be tracked as a function of broad-scale social factors – such as changing norms and values. The ABM framework presented in the paper does not even begin this process. The authors must think deeply about this, and improve the presentation if they want to sustain their claims in this context.

Overall I am very supportive of publication of the paper in HESS, eventually, but the presentation must be substantially improved, and some of the claims about contribution to socio-hydrology tempered.

---

## Referee Comment (RC2) · Anonymous Referee #2 · 23 Apr 2017

The title of the manuscript reminded me of the famous book by Epstein and Axtell (1998): 'Growing Artificial Societies: Social Science from the Bottom Up'. To my surprise, no reference was made to this relevant work. Even more surprisingly, the manuscript does not provide an adequate literature review on the considerable amount of literature on empirical agent-based or geo-simulation models from the fields of land-use change, water resources management and the modelling of complex socio-ecological systems in general (eg. An 2012; Torrens 2010; Filatova et al. 2013).

However, my main concern is that the paper is not focused. The objective of the study is not well described. What are the emergent phenomena that you wanted to study? The objective as described In the Discussion section (p12 line 8:'...to explore the essential

elements of agent-based modelling in a simple socio-ecological context...') was not well addressed by the manuscript.

Model description is poor. I think that an ODD description (Grimm et al., 2010) would have helped here. The model is also not available on OpenABM (www.openabm.org). This would allow reviewers, and others interested, to have a look at the model.

From what is presented with regard to the model I learned that it is not related to any empirical case-study and the comparisons between the 'real game' and the model are not very informative either. I found the text in the Discussion and Conclusions sections on validation and 'proof of concept' therefore not convincing.

References An, L. (2012). Modeling human decisions in coupled human and natural systems: Review of agent-based models. Ecological Modelling, 229, 25-36 Filatova, T., Verburg, P. H., Parker, D. C., & Stannard, C. A. (2013). Spatial agent-based models for socio-ecological systems: Challenges and prospects. Environmental Modelling and Software, 45, 1-7 Grimm, V., Berger, U., DeAngelis, D. L., Polhill, J. G., Giske, J., & Railsback, S. F. (2010). The ODD protocol: A review and first update. Ecological Modelling, 221(23), 2760-2768 Torrens, P. M. (2010). Agent-based Models and the Spatial Sciences. Geography Compass, 4(5), 428-448

---

## Referee Comment (RC3) · Anonymous Referee #3 · 25 Apr 2017

The paper is on the presentation of a "signal-based" instead of a "process-based" agent-based modeling framework for "the existing Irrigation Management Game".

Main issue:

The discussion needs some links to the findings of the previous studies of IMG. For instance are the stylized facts in Figure 5 (amount of soya depends on the position along the river) consistent with the previous findings of IMG. The boxplots of figure 6 are meaningless if not compared with previous IMG studies. How should the ms provide a proof of concept if model outcomes are not compared with the original game? How sensitive are the findings to changes in the parameters?

Overall I like the study but the presentation of the "concept of proof" of the framework to capture key elements of the IMG is immature at this stage. I think it will be crucial to identify stylized facts or patterns from the IMG literature and reproduce them with the ABM.

Minor comments:

Personally I think the Modelling Philosophy paragraph does not add much to the ms and distracts the reader.

The paragraph 2.2 on "the basis" of the irrigation game is important. The work by Marco Janssen is only cited in the text, but has not made it in the reference list (e.g. Janssen et al. 2012, Janssen et al. 2013,. . .). The authors should also consider Perez et al. 2016, Global Environmental Change – Human Policy Dimensions. Figure 1 is a snapshot from the original publication? It is almost unreadable and should be replaced by a more informative and aggregated graphical visualization that more easily explains the setup.

Paragraph 2.3: "Intra-farmer interactions" -> "Inter-farmer interactions" Personally I am not overly convinced by the combinatorial arguments for the game simplification. The research questions and the context should be more important of course it has to remain computational feasible. But why not reducing the number of players? Fig. 3 could be skipped.

Paragraph 3.1 is out of place it needs to go before analysis.

---

## Referee Comment (RC4) · Anonymous Referee #4 · 5 May 2017

The manuscript by Bouziotas and Ertsen presents a simple agent-based model for applications in irrigation systems. The authors state that the objective of this manuscript is to provide a proof of concept of the applicability of such approaches to the specific case of irrigation management.

The topic is definitely of interest for the HESS readership. Yet, the manuscript lacks of focus and, under a certain point of view, does not deliver what it promises. All in all it reads more as a book chapter than as a research paper. In particular, the novelty of the approach with respect to the existing Irrigation Management Game remains unclear – in several occasions, the authors write 'as in the original version of IMG'. The authors refer to the 'agent perspective' (P 2, L20), but it eluded me how this perspective is really

included in the model (particularly considering that all the agents in the model behave in the same way in the model application presented here). The results are somewhat expected given the rules of the game (upstream farmers have more choices and hence end up preferring more valuable crops, with positive effects on their incomes). I would also like to point out that several points are made regarding possible future works and analyses based on the proposed approach, not only in the discussion but also when presenting the approach. Taking up at least some of them could be a way to enhance the impact of this work and also to make it more focused on specific issues, should some issues be chosen to explore a specific problem or set of problems. There are several references to 'real irrigation systems', yet none of these aspects are explored.

Several sections could be significantly shortened without reducing the information content (Section 2.1 for example) and there are several concepts reiterated in the manuscript. At the same time, some parts need more explanations. Examples are i) the different roles of Manager and Gate Controller, which is not immediately obvious; ii) what is meant with m=3 on P5 or 'threshold of farmers' on P6 L3; iii) the physical or physiological meaning of 'good' or 'medium' conditions on P8.

Minor comments: - Not all the references cited in the text are reported in the reference list.

- It should be clear already at the beginning of section 2.3 there will be three cases explored.

- Why are all agents refer to as males?

---

## Author Comment (AC1) · 11 Jun 2017

We would like to thank the reviewer for the supportive attitude towards our paper. We appreciate the constructive criticism and remarks on the purpose, clarity and contribution of this work. What we hope to contribute to socio-hydrology in a fundamental way is a perspective not commonly seen in modeling studies: a modeling philosophy plus proof of concept in which social agency play a major role and a model that is built upon the agents' signal-based perceptions of (and actions towards) the hydrological reality. This was demonstrated precisely by constructing an ABM based on signals – which we detail further in our general comments. The Irrigation Management Game, from which

the ABM was based, provided a serious gaming application to a particular realm that is directly related to real-word irrigation systems, given in a level of abstraction and simplicity that helps the readers understand the core ABM mechanics without being lost in too much real-word complexity. At the same time, the IMG is an environment in which agent actions, decisions and experience feedbacks form the heart of the gaming experience, thus emphasizing the social aspects within a socio-hydrological system. We therefore assessed that this could be a prime case where our bottom-up, agent-based proof of concept could be based upon.

The purpose of our signal-based ABM was to develop a modelling methodology that allows to study agent actions, their possible effects for other agents, and possible results in terms of water distribution, crop growth and wealth creation – as these are parameters relevant for our immediate serious game environment and for socio-hydrological systems in general.

We agree that the emerging (up-scaled) effects from our ABM are not so much found in a community response. We do observe emergent effects though, as we show that within the IMG-ABM the series of decisions that are made by the agents create global patterns well-known in gravity irrigation in general and the IMG in particular. The general dynamics that were discussed in previous settings are apparent; upstream users generate more financial revenue and use more water, whereas downstream users generate less revenue, but generally more revenue per unit of water. We agree, however, that some of the claims on emergence need to be nuanced, and we aim to clarify our position on and examples of emerging effects in an updated version of our work.

We are not certain about the reviewer's comment that we should highlight the "social" factors of the socio-hydrology of our system. As described above, we believe the modelling philosophy and agent-oriented, signal-based logic offers a significantly improved view and schematization of social interactions compared to many previous works. We have given focus on human actions in a water system, which create new properties of the water system itself; the cascading effects seen in the system output

and other agents downstream is an important social and socio-hydrological mechanism. It is indeed correct that we did not include elements or properties of higher-order social relations yet, which may have created the feeling of underwhelming model results. However, we wanted to discuss our signal-based concept and general approach first before moving to further refinements. Moreover, we feel that more complex social interactions, such as scenarios of changing norms and values, will over-parameterize a simple modelling layout at this proof-of-concept stage. We believe that adhering to simplicity at this stage is needed, since the coupling with the real IMG is loose (i.e. the real gaming setting is not optimized to provide a detailed database to study, develop and calibrate the digital template) and thus we can only observe general mechanisms and dynamics and record simple agent interaction from real game settings. We believe a stronger coupling with an updated version of the real gaming environment could indeed allow the study of higher level social interactions between agents. If desirable, we can include our elaboration and ideas on how that could be done – and the evident social data needs from future versions of the IMG.

We do not think that our critique of lumped, top-down approaches can be easily met once we have additional and more appropriate observations. This is indeed closely related to a philosophical point of view of how societies and social relations are built and how they can be studied. We have not included much on this, as our position is discussed in other papers (especially by the second author), but we could clarify our position better in this HESS contribution.

Given these comments, we do not agree that this work pays lip service to how ABM can increase our understanding within socio-hydrology. We would argue that our first modelling efforts represent a rather important different perspective to building socio-hydrological models, where human agency is modeled at the individual scale and social interactions are more clear – and thus, better mapped and modeled - than in the case of observing and modeling lumped societal behaviour. We agree that we provide limited results, but would argue that these results – the core dynamics of irrigation networks,

reflected through a simple structure - provide clear evidence of the promising nature of our concept. It is obvious that we need to improve the presentation of and focus to our argument – for which we further propose ideas, not only in this answer but also to our general comments.

———————————————

---

## Author Comment (AC2) · 11 Jun 2017

The reference of the reviewer to Epstein and Axtell is highly appreciated, as it is indeed a useful base for the type of work we do and the argument we would like to build. Similar to what we do, these authors discuss computational frameworks as laboratories for exploring micro-mechanisms that may or may not generate social phenomena. They show that simple local rules often are enough to create more complex patterns or overarching effects. Our IMG-ABM is also based on local rules, and results in emerging global patterns, shaped by simple agent action which depends on model settings. We are indeed aware of the considerable amount of literature on empirical agent-based or

geo-simulation models in many fields. We could have included many more indeed and think that our current selection is rather well-balanced, as this study does not aim to provide an extensive literature review on ABM. Having said that, we will certainly profit from the reviewer suggestions and will use it for our revision, as stated in our general comments as well.

As this reviewer is also asking about how we study emergent phenomena, our response to reviewer 1 is also relevant here. The emerging (up-scaled) effects from our ABM are not so much found in responses from social institutions at this stage. We do observe emergent effects though, as we show that within the IMG-ABM the series of decisions that are made create patterns well-known in gravity irrigation in general and the IMG in particular. Upstream users generate more financial revenue and use more water, whereas downstream users generate less revenue, but generally more revenue per unit of water. We aim to clarify our position on and examples of the emerging effects seen in our work.

As explained in our paper, our model is based on the empirical case-study of the Irrigation Management Game. We compare the "game as played by humans" and "game-based model". An elaboration of the game results from humans and the reasons why the IMG would count as "real" have been published elsewhere, but we can include more details in (and make a comparison with) our current work.

What we aimed for was to discuss "essential elements of agent-based modelling in a simple socio-ecological context", in terms of agent properties, signal logic and relevant parameters that can be calibrated based on the real setting. It is clear, in line with the comments from other reviewers, that we need to clarify our modelling description and will, among other improvements, consider using the ODD system suggested. Reflecting our belief in scientific openness also seen through our submission in HESS, we planned to make the model publicly available (e.g. through OpenABM) when our paper would be published.

---

## Author Comment (AC3) · 11 Jun 2017

We thank the reviewer for the positive assessment of our study. Similarly to our answer to Reviewer #2 on relating our model findings to results of previous uses and studies of the IMG, we stress that actual game results from humans and the reasons why the IMG would count as "real" have been published elsewhere. We will, however, provide clarified links and include more details in the current paper. We would like to refer to our general comments for more clarity on how we aim to improve the presentation of our proof of concept to capture key elements of the IMG. We thank the reviewer for the detailed additional minor comments, which we will work on.

---

## Author Comment (AC4) · 11 Jun 2017

We appreciate the clear criticism and useful remarks on the clarity of our work. The reviewer starts by raising concerns on the paper focus, which we acknowledge and discuss about in the general comments, as well as remarks about the connection to the real Irrigation Management Game (IMG). We would like to point out that the relation between the existing real game setting and our model is twofold. First, the IMG serves as a realistic, yet artificial case reality to set the model. Second, our model players are compared with the human players in the original game – and the results of their actions as well. The 'agent perspective' is included in our model through the choices

that agents can make – and indeed at the moment they make similar choices.

Regarding the expectancy of the results, we agree that most of the results are not terribly surprising, as we follow the original rules of the game – including allowing upstream farmers to take water first. We have not done so yet, but could easily implement a water distribution logic where downstream farmers can start first. We note, however, that this does not reflect the original rules of the game, which represents a simple gravity system without elaborate management policies, where upstream users have a clear advantage, as is the case for many real-world cases especially in the developing world. Moreover, we would like to note that not all the results are straightforward effects coming from the game rules. For instance, what is not directly modeled but appears in the results is the crop-per-drop 'wisdom' that downstream users develop during the game.

We believe that what is crucial for our current paper is how one can understand the model – and as such the IMG and, in turn, real world irrigation – as a small world in which agents relate to other agents and the environment through signals. Obviously, we have modeled how signals change from upstream to downstream, but we could have done otherwise. This brings us to the points we made regarding possible future works and analyses. We have some proposals how to strengthen these claims in our general comments.

We thank the reviewer for the minor suggestions for shortening and expanding elements in the text and will work on these. We have indeed referred to all agents as males, for no better reason that in the second author's language all words that are not clearly female are to be addressed as being male. However, we see the bias in our reference of male agents, and will make changes to accommodate female agents as well.

---

## Author Comment (AC5) · 11 Jun 2017

We would like to thank all reviewers for their supportive and critical comments on our paper. We note that several topics, such as the lack of certain literature or issues on presentation clarity arise in multiple comments. We are thankful for the correlated feedback and would like to highlight the following changes that we aim to do to improve our work:

- Several authors raise concerns that parts of the ABM literature are missing. We aim to do a careful review of our literature section, adding useful work pro-

posed by reviewers where necessary. Likewise, we aim to stress the function-ality of ABM where society is an important subsystem, always having in mind not overemphasize theoretical work over our practical point of view, which is an application in irrigation modelling.

• We acknowledge that the presentation of our argument needs to be improved. Besides the building blocks in our answers to the individual reviewers, we aim to improve the paper layout by revising Section 2, so that the notions of general modelling discussed are more relevant to the case study and less distracting to the reader who follows our argumentation. Moreover, following the remarks of a number of reviewers, we aim to revise the aesthetics of the contained figures by redrawing a number of them, which will help clarify information to the reader. We aim to link figures related to our results with patterns previously seen in the IMG.

• Likewise, we would like to make changes in the text so that: (a.) The emergent effects that we expect to see in irrigation systems are clarified and discussed fur-ther, (b.) the contribution we expect to have from our work is nuanced, having in mind the limitations of our proof-of-concept level, (c.) the purpose and limitations of our work are made more direct.

• Finally, we aim to clarify the links with the real IMG, both in the methodology but also in the results and patterns observed. Our work will be more explicitly linked with previous results and real applications of the IMG.

We end our response by providing a summary of what we tried to do in our paper, with some additions and rephrasing to reflect the proposed changes. We kindly ask the editor to comment on whether our points below can be used as main considerations to revise our paper.

• As Agent-Based Models (ABM) are one of the most promising choices to include human agency as an equally important part in coupled human-water systems

compared to the water system, we have created an ABM setting using the main features of a real serious game setting, the Irrigation Management Game (IMG). The irrigation system we model is man-made, as irrigation systems are, but represents socio-hydrological realities as irrigation typically modifies existing hydrological realities.

• We have modelled the game's unfolding as a procedure of interaction and communication between farmers and the water agency, as well as between farmers themselves. The human agents act according to defined rules, and the resulting simulation provides results in terms of farmers' water use and financial revenue as well as system's use and revenue.

• Our ABM is a model-based methodology built at the agent scale, allowing the study of agent actions, possible effects for other human agents, and possible results in terms of water distribution, crop growth and wealth creation. The results can be then aggregated to view the response of the whole system.

• We show that within the IMG-ABM the series of decisions that are made/modelled create patterns well-known in gravity irrigation in general and the IMG in particular. Upstream users generate more financial revenue and use more water, whereas downstream users generate less revenue, but generally more revenue per unit of water.

• We do realize that our paper does not offer a complete recipe for agent-based applications in real irrigation systems. However, we find the simplified setting we presented a satisfactory case study to highlight a bottom-up, agent-based modelling philosophy of socio-hydrological systems to the reader. We want to explore essential elements of agent-based modelling in socio-hydrology – using irrigation as a suitable modelling context, given that irrigation is a socio-hydrological entity.

• Using the IMG, with ample data available from multiple applications, allows

checking whether our model results are realistic. As such, we can explore ways for integrating (at least) human agency in socio-hydrological models without issues of data scarcity or parameterizations heavily dependent on qualitative variables like emotion and trust. The model parameters represent tangible probabilities reflecting agent choices, which can be in turn measured in an adjusted real IMG setting.

- Based on our modelling efforts, our discussion on how to define future studies is also suggesting a research agenda for socio-hydrological bottom-up model building. The field may not need that many additional case studies, but an increased focus on actual model coding, parametrization and a seamless coupling between games with real game agents and the digital model.

- In order to study co-evolutionary transformations of water-related practices and arrangements, we propose that we should model socio-hydrology as social practices that typically include human agents and material objects. We conceptualize these practices as actor-networks, in which (sets of) actions are employed to realize goals and conditions.

- What we claim is that the agent-based template we provide offers a solid base for bottom-up socio-hydrological modelling of such practices. Despite the applied simplicity, we have been able to produce emergent dynamics that are not predefined in what agents actually should do. In other words, we have local model agents that make "blind", local decisions and yet produce clear upstream-downstream patterns.

- Our template can be used as exploration and basis for more elaborate human agency in socio-hydrological cases – with irrigation systems being good examples as we have good examples and good control of material options and boundaries. The next step should be to add complexity, especially in agent decision-making and cooperation.

[Figure]

- Agents process information continuously; this sequence can be synthesized as "action - result - judgement/perception - (re)action". When models offer agents a spectrum of possible actions, agents can select possible actions based on perception(s) of the actual situation. The advantage is that perceptions can be changed in the analysis, with actions and results that have defined physical boundaries. This allows validation of model results.

- Our presented work is clearly not including it yet, but we would like to suggest that our template offers one advantage in terms of modelling human-water coupled systems: we can treat human and water agents in equal modelling terms. Both agents are able to act – in model terms. In our irrigation setting, canal hydraulics are key, given the many observations in actual irrigation systems that show the importance of flow regimes and their actual behavior. As we mention in the paper as well, such additional detail can be readily implemented in the model setup without deviations from the logic of signals our model is based on. However, one should be cautious that this architecture needs a data space that is not provided by the current version of the real game setting, so adjustments need to be made there as well.

- We consider water as signal that moves through the model environment. Agents receive signals, appreciate signals, act upon signals and release (other or the same) signals. Within this logic, there is no need to appreciate human agents differently from non-human ones. A "Canal Agent" accepts water signals from upstream positions and re-shapes them, based on hydraulics, for downstream human and non-human agents. We argue that this conceptualization in agent-based models allows examining how human and environmental agents together modified irrigated landscapes – or socio-hydrology in general.

- Our action-oriented, signal-based modelling methodology can offer a spectrum of possible actions to all model agents. Obviously, that does not mean that all

agents judge in the same way. Human agents may decide to trade, material agents may 'decide' to change their value. Given the many choices available for all agents, and the restrictions we can impose on how model agents update their status, we will have an ensemble of outcomes – or many likely outcomes. We will also have outcomes which are not feasible in terms of physical boundaries being reached or observed.

- As such, our signal-based template brings the timescales of studies of humans and hydrology together. Complex interactions between (human and non-human) agencies are produced at the same temporal and spatial scale – with possible emerging properties at larger spatial scales or later in time – similar to our revenue patterns or water distribution. Larger hydrological scales, for instance changes in catchments or the hydrological input, can be readily modeled as the simulation times that are presented are long.

- Our main aim is to discuss how our mechanics of constructing an ABM based on signals helps socio-hydrology as a field of enquiry. Our IMG is a computational laboratory to explore micro-mechanisms in water systems that may or may not generate social phenomena. Simple local rules appear to be enough to create more complex patterns or overarching effects – whether creation is done by humans or non-humans. We firmly believe that these concepts will aid socio-hydrology in general, as a valuable modelling supplement to the top-down philosophy prevalent in previous hydrological applications.